# Bone Loss in Implants Placed at Subcrestal and Crestal Level: A Systematic Review and Meta-Analysis

**DOI:** 10.3390/ma12010154

**Published:** 2019-01-05

**Authors:** Natalia Palacios-Garzón, Eugenio Velasco-Ortega, José López-López

**Affiliations:** 1Department of Odontostomatology, University of Barcelona, l’Hospitalet de Llobregat, 08907 Barcelona, Spain; npalaciosgarzon@gmail.com; 2Faculty of Dentistry, University of Seville, 41009 Seville, Spain; evelasco@us.es; 3Department of Odontostomatology, Medicine and Health Sciences (School of Dentistry), University of Barcelona, 08907 Barcelona, Spain

**Keywords:** systematic review, subcrestal, crestal, bone loss, implants, meta-analysis

## Abstract

Background: To assess differences in marginal bone loss in implants placed at subcrestal versus crestal level. Methods: An electronic and a manual research of articles written in English from Jaunary 2010 to January 2018 was performed by two independent reviewers. Clinical trials comparing bone loss for implants placed at crestal and subcrestal level were included. Pooled estimates from comparable studies were analyzed using a continuous random-effects model meta-analysis with the objective of assessing differences in crestal bone loss between the two vertical positions. Results: 16 studies were included; 10 studies did not encounter statistically significant differences between the two groups with respect to bone loss. Three articles found greater bone loss in subcrestal implants; while 3 found more bone loss in crestal implants. A meta-analysis for randomized control trial (RCT) studies reported an average and non-statistically different crestal bone loss of 0.028 mm. Conclusions: A high survival rate and a comparable bone loss was obtained both for crestal and subcrestal implants’ placement. Quantitative analysis considering a homogenous sample confirms that both vertical positions are equally valid in terms of perimplant bone loss. However, with respect to soft tissue; in presence of a thin tissue; a subcrestal placement of the implant should be preferred as it may reduce the probability for the implant to become exposed in the future and thus avoid the risk of suffering from peri-implant pathologies.

## 1. Introduction

Dental implants have become the preferred choice for the replacement of missing teeth. The five-year success rate of dental implants has increased from 93.5% to 97.1% within the past decade, with a higher survival and a lower complication rate [1]. Patients increasingly require treatments that offer more aesthetics and comfort, making implantology a demanding field, where, obtaining osseointegration or meeting the success criteria of implants highlighted by Buser et al. in the 1990s [2], such as lack of pain and infection, absence of radiolucency and mobility and possibility of restoration, is no longer considered a sufficient condition.

Research in the area of implantology has been evolving substantively. Scientists begin to devote their attention to physical and chemical properties of the implants; on creating different types of surfaces and degrees of roughness [3,4] with the objective of reducing the healing time to achieve secondary stability [4,5]. Furthermore, researchers have also focused, among others aspects, on finding the most effective and safest connection between the implant-abutment, the best geometric design [4] and the most favorable placement of implants in relation to the crestal bone [6].

Despite the effort in research and the advancement in implant design, when an implant is placed, it is assumed that there will be an unpredictable loss of bone around it, difficult to perfectly forecast beforehand. In 1986, Albrektsson et al., published a seminal paper, which is still a reference today, where it is argued that a bone loss of less than 1.5 mm during the first year after implant placement and less of 0.2 mm annually in the following years, can be considered satisfactory [7].

According to the current literature, the preservation of crestal peri-implant bone is considered a key feature for the success of the treatment given that the bone around the implant determines the stability of the soft tissue, which in turn is a crucial aspect for esthetics and long-term survival [8,9].

Within these lines of research, several authors describe that the position in which the implant is placed with respect to the crestal bone, is a fundamental factor to preserve the bone in the future [10,11,12,13,14]. Although, with respect to this issue there is currently a controversy. Some authors recommend placing the implant under the crest of the bone (i.e., subcrestal placement). They argue that this specific position would contribute to the preservation of the mucosa [10], helping to obtain an ideal emergence profile in esthetic areas [11,12] and would prevent the surface of the implant from being exposed while reducing the likelihood of suffering from mucositis or peri-implantitis [13]. Along the same lines, studies with animals pointed out that implants placed at subcrestal level were characterized by less bone loss with respect to implants placed at the level of the crest [10,14].

By contrast, other authors reported evidences highlighting an increase in bone loss for implants placed subcrestally [6]. A possible reason for thiscould be attributed to the bacterial colonization of the implant-abutment junction, where an inflammatory infiltrate is produced [15,16,17]. This feature, in concomitance with a low concentration of oxygen, could create an ideal ecosystem for the proliferation of anaerobic bacteria [18]. Moreover, according to de Siqueira et al. (2017), it could also be speculated that the deeper placement of the implant may correlate with embolism, deeper pocketing and inflammation of the soft tissue [12].

Given the controversial results encountered in the literature, the objective of this review is to systematically evaluate the influence of crestal and subcrestal placement implants with respect to crestal bone loss and soft tissue and as a result shed more light over this important question.

## 2. Materials and Methods

The literature considered for this systematic review is based on the PRISMA’ (Preferred Reporting Items for Systematic Review) [19] guidelines and aims at answering the following specific question built on the PICO (Population, Intervention, Control, Outcomes) format [20]. For patients treated with dental implants, are there differences between subcrestal and crestal implants placement with respect to marginal bone loss?

(P) Population: Patients treated with dental implants.

(I) Intervention: Implant placement at the subcrestal bone level.

(C) Control: Implant placement at the crestal bone level.

(O) Outcome: Bone loss around implants placed at subcrestal and crestal level.

### 2.1. Inclusion and Exclusion Criteria

Inclusion criteria: recent human clinical studies, comparing bone loss in implants placed with different vertical positions, at crestal level and subcrestal level. A follow-up period of at least one month in implants placed in one stage and six months in implants placed in two stages.

Exclusion criteria: articles published in languages other than English. Articles analyzing uniquely a type of placement (crestal or subcrestal). Articles published before 2010.

### 2.2. Search Strategy and Selection of Studies

An electronic search was carried out in PubMed/MEDLINE. The terms used in this search were: “subcrestal placement and crestal placement implants”, “bone loss in implants placed crestal-subcrestal”, “subcrestal implants crestal implants”. A manual search was carried out on the obtained articles, in order to find more articles meeting the inclusion criteria.

Two reviewers (N.P.G. and J.L.L.) assessed independently all titles and abstracts obtained from the electronic search in order to reach a consensus on the decision to exclude or admit each study.

### 2.3. Data Extraction

Given the high degree of heterogeneity characterizing the studies included in the review, a meticulous analysis of the data was carried out in order to compare them. We extracted information about patients (age, gender, number of implants placed, number of patients treated), characteristics of the implants and surgical techniques adopted; whether the abutment was placed in a second procedure (two stages/submerged) versus implants in which the abutment was placed immediately (one stage/non-submerged); loading of the implants: delayed versus immediate loading; the type of prosthesis, timing of implant placement (alveolar socket healing versus immediate post extraction); and follow-up period, the survival rate of the implants and bone loss. Regarding the soft tissue, different parameters were analyzed: probing pocket depth, bleeding on probing, modified plaque index, modified gingival index, keratinized tissue, vertical mucosa thickness and histological analysis.

### 2.4. Methodological Quality of Each Study

Each study was evaluated using the Jadad scale [21], this evaluation method consists of assessing the methodological quality of the clinical trials. The score goes from 0 to 5, being 0–2 of low quality, 3 of medium quality and 4–5 of high quality.

### 2.5. Statistical Analysis

Pooled estimates from the studies were analyzed using a continuous random-effects model meta-analysis. The variable analyzed was crestal bone loss of implants placed at a crestal and at a subcrestal level. Forest plots were produced to represent graphically the difference in outcomes of crestal bone loss. *p*-value = 0.05 was chosen to determine whether differences were statistical significant. Heterogeneity was assessed with χ² test and I2 test. R version 3.3.2 (Foundation for Statistical Computing, Vienna, Austria), and R studio version 1.0.44 Studio, Inc, (Boston, MA, USA) were employed in the statistical analysis.

## 3. Results

In the electronic search, a total of 150 articles were identified. Using the keywords “subcrestal placement and crestal placement implants”, 44 items were found, of which only 11 met the inclusion criteria. With the keywords “bone loss in implants placed crestal–subcrestal”, 39 items were found, and only 2 met the inclusion criteria. With “subcrestal implants crestal implants”, 67 studies were obtained and 0 studies met the inclusion criteria. Three more articles that met the inclusion criteria were found in a manual search in the literature and were included.

Finally, 16 studies were included in this review for the qualitative synthesis [6,12,22,23,24,25,26,27,28,29,30,31,32,33,34,35]. Nine were randomized control trials (RCTs) and the other 7 were non-randomized control trials (non-RCT). For the quantitative synthesis regarding bone loss, 5 RCTs articles were included [6,28,29,32,35] (Figure 1).

### 3.1. Methodological Quality Assessment

The methodological analysis of Jadad (Table 1) shows that 8 RCT presented a high scientific quality with scores of 4 and 5, except one of medium quality with a score of 3. Of the 7 non-RCT, 6 were of low quality with scores between 1 and 2 and one of medium quality with a score of 3.

### 3.2. Characteristics of Studies Included

Of 9 RCTs, 5 did not find significant differences in bone loss [12,24,26,27,32], two articles found a higher loss of bone in implants placed at the subcrestal level [6,29] and 2 articles found less bone loss in implants placed at the subcrestal level [28,35].

Of 7 non-RCTs, 5 did not found significant differences between crestal and subcrestal implants [22,23,25,30,31]. One study found that subcrestal implants presented less bone loss, though statistical significance was not reached [33] and one study found grater bone loss in subcrestal implants [34].

The majority of the studies analyzed uniquely the crestal and subcrestal positions, except for 4 studies that in addition analyzed the supracrestal position [22,30,32,34]. A total of 1346 implants were placed, of which, 1093 were included in this review and the remaining 253 were excluded because they were placed in the supracrestal position. Of 1093 implants included, 604 implants were placed subcrestally, and 489 were placed at the crestal level. Fourteen articles reported the age of the participants, which ranged from 23 to 82 years, the other two articles did not report it [31,33]. The number of patients treated varied between 9 and 85. Of 16 articles, 10 (62.5%) did not obtain significant differences between crestal or subcrestal implants in terms of bone loss. For the remaining 6 articles (37.5%), 3 found greater bone loss in implants placed at subcrestal level, while the other 3 remaining articles found the opposite, more bone loss in the implants placed at the crestal level. Of 1093 implants, 698 were placed in two stages, 302 in one stage and 93 were immediate loading. The subcrestal implants were installed in a depth ranging from −0.5 mm to −3.4 mm apical to the alveolar ridge and the crestal implants were placed between 0.0 mm and a maximum of 0.75 mm above the level of the crestal bone. With respect to the timing of implant placement, in a study with a sample of 20 implants, all were placed post-extraction [27], 6 studies declared that the implants were placed on healed edentouls ridges. In 3 studies implants were placed at least 3 months after the healing of the dental alveolus. In 1 article, implants were placed 6 months after extraction while 6 studies did not report this information. Of the 16 studies, only 3 performed immediate loading, of these 3, two found no significant differences [12,26] and one found greater loss in implants placed at the crestal level [33]. Of the 13 studies with delayed loading, 8 did not find significant differences [22,23,24,25,27,30,31,32], 2 found greater loss in implants placed at crestal level [28,35] and 3 found greater loss in implants placed at subcrestal level [6,29,34] (Table 2a,b).

The average survival rate was particularly high and equal to 99.4%, representing 629 implants for the studies reporting it. Four studies, representing 464 implants, did not report the survival rate [23,25,31,34].

In the RCT studies the follow-up of implants placed varied from 3 to 36 months while in the non-RCT it ranged from 1 to 155.35 months (Table 2a,b).

### 3.3. Characteristics of the Implants

Six articles described the type of implants’ surface used [12,22,27,33,34,35], the rest did not report it. Regarding the length of the implants, this ranged from 8 to 14 mm and was reported in 6 studies [6,23,24,28,29,32], the diameters varied from 3.3 to 5 mm and was reported in 8 studies [6,23,24,27,28,29,32,35] (Table 3).

With respect to the implants’ connection, in 11 studies implants were characterized by internal connection [6,12,22,23,24,26,27,28,29,33,34,35], 8 of them using a Morse taper connection [12,22,24,26,27,28,29,33]. Two studies did not report the type of connection used [25,30]. One study used external connection [31] and one study combined external connection with internal connection both for implants placed at crestal and subcrestal level [34]. In the rest of the studies, the implants used had the same characteristics. The implant-related characteristics of studies are reported in Table 3.

### 3.4. Bone Loss

Among the 16 studies considered, in 10 there were no significant differences between the two groups with respect to bone loss. In 3 studies, one non-RCT [34] and two RCTs [6,29] with similar characteristics among them, with the exception of platform-switched that was not used in the non-RCT study, the bone loss was significantly higher for subcrestal implants. These implants were placed in two stages, with a follow-up of between 24 and 36 months. In the remaining 3 articles, 2 RCTs [28,35], and 1 non-RCT [33], the opposite occurred: Fickl et al. (2010) [35] and Vervaeke et al. (2018) [28] with a follow up of 24 and 12 months respectively, both placed in one stage, subcrestal implants showed better bone levels than the implants placed at crestal level. In the remaining article, Degidi et al. (2011) [33], with a follow up of 2 months, the implants placed at subcrestal level found preexisting and newly formed bone over the implant shoulder. 

From the RCT studies, the highest amount of bone loss detected around subcrestal and crestal implants was exactly the same: 1.22 mm [6] and 1.22 mm [35] respectively.

Ten studies showed an average of bone loss of less than 1 mm in subcrestal implants and 11 studies a loss of less than 1 mm in crestal implants (Table 2a,b).

### 3.5. Quantitative Analysis in Bone Loss

A meta-analysis was performed to analyze the mean differences of crestal bone loss between RCT studies. Of the 9 RCT; five randomized studies [6,28,29,32,35] provided valid data to be included in the meta-analysis as they evaluated radiologically crestal bone loss of implants placed at crestal (n = 100) and at subcrestal level (n = 164) with a follow up period between 12 and 18 months. In addition, the five studies considered used the same type of implant connections (i.e., internal). Two studies were not included in the meta-analysis because the bone loss evaluation was not carried over with an intraoral/panoramic radiography. More precisely, a clinical evaluation with a probe was performed on Koh et al. (2011) [27] and a CBCT analysis was performed on Koutouzis et al. 2016 [26]. Additionally, the other two studies were not included in the meta-analysis due to the fact that they reported uniquely a follow-up period up to 8 months [12] and 3 months [24] respectively. The forest plot (Figure 2) shows a crestal bone loss mean difference of 0.028 mm and a *p*-value = 0.92 (95% CI (Confidence Interval): −0.591 to 0.648, heterogeneity I^2^ = 98.65%, *p* ≤ 0.001).

The negative values shown in the forest plot represent a higher crestal bone loss for the subcrestal group. While positive ones represent higher loss of bone for the crestal group.

With respect to the non-RCT studies, a meta-analysis could not be carried out due to the high level of heterogeneity between the studies included in the review. The main difference comes from the length of the follow-up periods which varies substantially in five out of seven studies: 1–2 months [33], 63 months [23], 6 months [25], 18 months [30] and 105 months [22]. Only Veis et al. (2010) [31] and Kim et al. (2017) [34] had comparable follow-up periods of 24 months and 36 months respectively. Nevertheless, in their case, the connection of the implants varied; being only external in one of the articles [31] and both external and internal in the other article [34]. Since differences in the connection may induce a bias in the result, we could not perform a meta-analysis for the non-RCT studies.

### 3.6. Soft Tissue

Regarding the soft tissue, in 4 studies no significant differences were found between both vertical positions [12,23,29,32]. Koh et al. (2011) [27], found better results for the subcrestal implants in terms of keratinized tissue width, the difference being statistically significant. Palaska et al. (2016) [24], found a significant and highest values in the modified gingival index for the crestal group.

Degidi et al. (2011) [33], were the only ones who performed a histological analysis, the implants were retrieved from patients two months after their placement. In implants placed at the crestal level, dense connective tissue was found in the coronal area of the implant, while in subcrestal implants no gaps or fibrous connective tissue were found at the interface of the implant.

Vervaeke et al. (2018) [28], found a significant correlation between soft tissue thickness and bone level alterations after 6 months, with higher bone loss for crestal implants when thin tissue was present. 

In 8 studies considered in this review, soft tissue was not analyzed (Table 4).

## 4. Discussion

As described in the result section, in 10 studies there were no statistically significant differences between implants placed at crestal and subcrestal level with respect to bone loss around them. Three studies found a greater loss in implants placed at the subcrestal level and the 3 remaining studies found the opposite, a better preservation of the bone in the subcrestal implants. According to the vast majority of these studies, there is no difference of bone loss between implants placed at crestal or subcrestal level. Furthermore, as seen in our meta-analysis with a selected homogenous sample, a crestal bone loss occurs in both cases, and the difference is not statistically significant.

If we take into account that every time we place an implant we expect a loss of bone that can cause the implant exposure, aesthetic problems and the risk of suffering perimplant pathologies, it seems natural to think that positioning the implant deeper with respect to the bone crest may be favorable in order to avoid these potential drawbacks.

In the last few decades, researchers devote substantial effort to shed more light over the causes responsible for the small loss of bone occurring when an implant is placed especially after the first months following the placement. As pointed out in the results of this review, the vertical position of the implant with respect to the bone does not seem to be the main cause of bone loss. This conclusion is as well in line with the results obtained in Gualini et al. (2017) [36], where no significant differences regarding bone loss were found for implants placed in different subcrestal positions. Instead, we might speculate that the loss of bone could differ if we take into account the interaction of more than one factor simultaneously in addition to the vertical position of the implant such as the platform switching, the type of connection, the soft tissue characteristics, or the biological width. In the following, we will analyze the impact of each one of these factors on bone loss with respect to the crestal and subcrestal placing of the implant.

### 4.1. Platform-Switching in Relation to the Vertical Position of the Implant

In the concept of platform-switching, the implant/abutment junction moves towards the center of the implant, with the aim of separating the bacterial filtration from the crestal bone [37]. This type of design, thanks to the narrowing of the implant/abutment junction may help to minimize the invasion of the biological width [33].

It is not clear whether platform switching can help to prevent bone loss. Within this review, 6 studies found differences between crestal and subcrestal implants, 5 of them used the concept of platform-switching and two of them found a greater bone loss in implants placed at the subcrestal level [6,29]. This result could possibly be attributed to a deeper implant placement that would be related to deeper pocketing and inflammation [6]. The other three studies [28,33,35] in contrast, found less bone loss in subcrestal implants when the platform-switching was used. Several authors advocate in favor of the platform switching based on the reduced bone loss observed as a result of its use [31,38,39]. Nevertheless, more research is needed with longer follow-up periods to determine if platform-switching is more beneficial in terms of reducing bone loss.

### 4.2. Connection in Relation to the Vertical Position of the Implant

With respect to the implants’ connection, almost the totality of the studies considered in this review used internal connection and only one study combined internal connection with external connection between implants placed at crestal and subcrestal level [34]. In this study they found a higher bone loss in implants placed at the subcrestal level both for internal and for external connections. This result is in line with a recent systematic review comparing internal and external connection and pointing out that there are not enough evidences to conclude that one connection is better than the other with respect to marginal bone loss [40]. Nevertheless, several clinical studies have suggested that the Morse taper connection with subcrestal implant could be favorable to prevent bone loss, being most efficient in terms of bacterial sealing and prosthetic stability [33,40].

### 4.3. Soft Tissue Response to Crestal and Subcrestal Implants

According with the results presented in the study of Verveake et al. (2018) [28], included in this review, in areas with thin soft tissue, we could anticipate bone remodeling if we adapt the position of the implant with a more subcrestal placement in relation to the existing soft tissue.

### 4.4. Biological Width in Relation to the Vertical Position of the Implant

The biological width is present in natural teeth and is composed of the epithelial junction and connective tissue. Abrahamsson et al., (1996) [41], demonstrated that this space is also formed around dental implants. According to Oh et al. (2002) [42], the bone remodeling of the crest is influenced by a variety of factors including the facilitation of the formation of the biological width around the implant with the objective of creating a barrier against the oral flora. The bone remodeling lasts one year after placing the implant, being more accentuated at the beginning, and it starts either the same day of the placement of the implant, if a pillar is placed immediately and the implant is exposed to the oral environment (one stage), or in a second moment if the pillar is placed later (two stages) [43]. It is for this reason that in our inclusion criteria we decided to accept studies with short follow-up periods, as long as the implants were placed mostly in one stage. The formation of biological width around implants could influence the bone remodeling and this fact might explain why the resorption is more pronounced during the first year, as reported in all the articles included in this review. Of the studies included in this review, Vervaeke et al. (2018) [28] is the only one that took into account bone remodeling and biological width, obtaining favorable results. In their study, the thickness of the soft tissue was measured before placing the implants and a non-random systematic assignment was applied to determine the position of the subcrestal implant while adapting it to the thickness of the soft tissue, leaving at least 3 mm of space for the establishment of the biological width. If a patient was characterized by having 2 mm of the width of the mucosa, they considered placing the implant 1 mm below the crest in order to anticipate bone remodeling and avoid future exposure of the implant.

As suggested by Kan et al. (2010) [44], the gingival biotype can be considered thin if the measurement is ≤1.0 mm and thick if it measures >1.0 mm. Several studies reported an increase in bone loss in implants placed at the crestal level with thin biotypes and have concluded that a minimum amount of keratinized tissue is needed around the implants to reduce tissue recession [45,46] and crestal bone loss [28,42,47,48,49].

## 5. Conclusions

The existing literature is not yet conclusive on whether one of these positions of implant placement is superior to the other. In this review we observed that in the vast majority of articles there are no differences and that the loss of bone that always occurs when an implant is placed, is not due strictly to the location of the implant with respect to the bone, as it continues to occur in both vertical positions.

Moreover, appropriate clinical results with a high survival rate and a similar bone loss were obtained both for crestal and subcrestal implants’ placement.

Although, with respect to soft tissue, in presence of a thin tissue, a subcrestal placement of the implant should be preferred as it may reduce the probability for the implant to get exposed in the future and thus, avoiding the risk of suffering from peri-implant pathologies. Nevertheless, to fully support this statement, more precisely designed and more homogenous clinical trials with larger samples and longer follow-up periods are needed.

## Figures and Tables

**Figure 1 materials-12-00154-f001:**
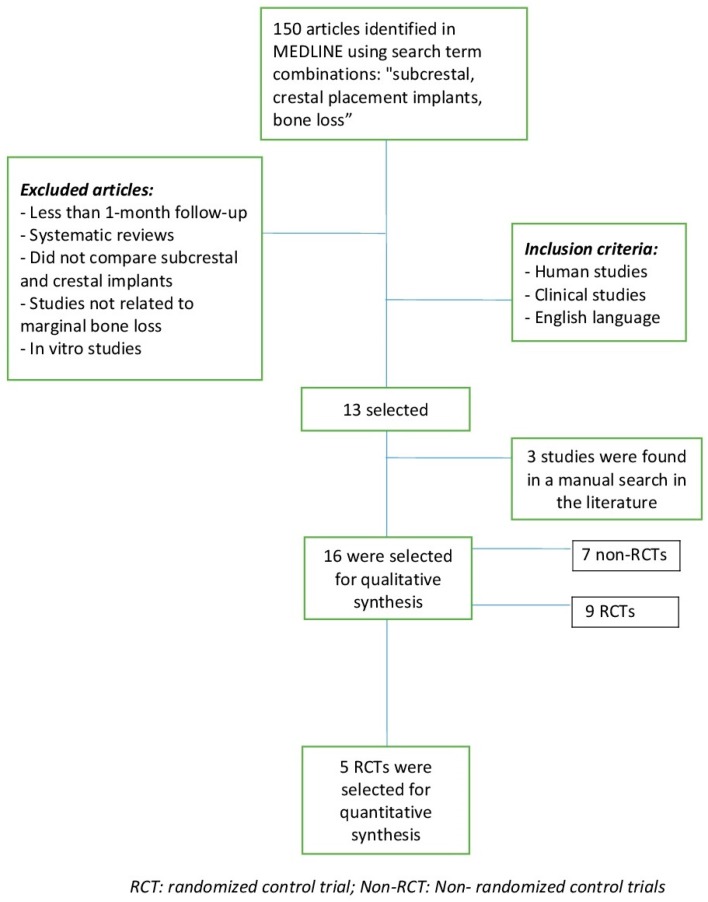
Flow chart.

**Figure 2 materials-12-00154-f002:**
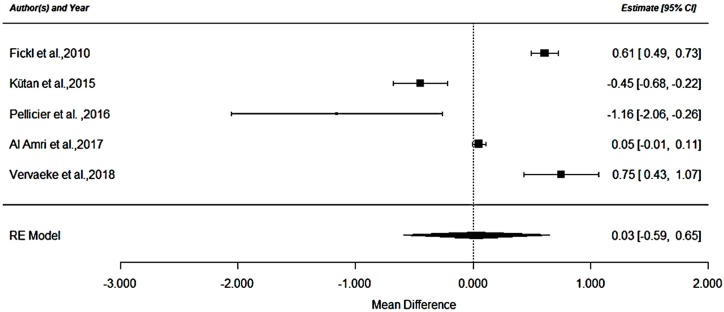
RCT analysis.

**Table 1 materials-12-00154-t001:** Jadad score in studies selected.

Questions Jadad	Q.1	Q.2	Q.3	Q.4	Q.5	Q.6	Q.7	TOTAL
Romanos et al. 2015 [22]	0	1: Implants placed by the same oral surgeon between 1993 and 2004	−1	1	1	0	0	2
Al Amri et al. 2017 [23]	0	1: Patients having undergone dental implant therapy for single missing tooth	−1	1	1	0	0	2
Pellicer et al. 2016 [6]	0	1: Using pre-defined randomization tables	0	1	1	0	1	4
Palaska et al. 2016 [24]	1	1: Using an online randomization plan generator	0	1	1	0	1	5
Nagarajan et al. 2015 [25]	0	0	−1	1	1	0‘	0	1
de Siqueira et al. 2017 [12]	1	1: A computer-generated random number table for patient allocation	0	1	1	0	1	5
Koutouzis et al. 2014 [26]	0	1: A computer-generated list to distribute the subjects. Treatment assignments were stored in sealed envelopes	0	1	1	0	1	4
Koh et al. 2011 [27]	0	1: Assigned by choosing a letter from a bag	0	1	1	0	1	4
Vervaeke et al. 2018 [28]	0	1: A systematic non-random assignment was applied to determine the position of test and control implants	0	1	1	0	1	4 *
Kütan et al. 2015 [29]	1	1: Was made by one of the authors by the flip of a coin	0	1	1	0	1	5
Ercoli et al. 2017 [30]	0	1: Patient had received a dental implant during a period of 6 years, from 2009 to 2015	−1	1	1	0	0	2
Veis et al. 2010 [31]	0	0	−1	1	1	0	0	1
Degidi et al. 2011 [33]	0	1: Search in the archives of the Implant Retrieval Center of the Dental School of the University of Chieti-Pescara, Chieti, Italy	−1	1	1	0	1	3
Al Amri et al. 2017 [32]	0	1: Randomization was performed by tossing a coin.	0	1	1	0	1	4
Kim et al. 2017 [34]	0	0	−1	1	1	0	1	2
Fickl et al. 2010 [35]	0	1: All implants placed between 1 January and 31 December 2006	−1	1	1	0	1	3

* Vervaeke et al. (2018) should have obtained 3 points as they did not use a random assignment of participants. Despite that, we decided to give them 4 points as the decision for a non-random procedure was linked to the sequential nature of their research objective. The Jadad scale of quality of the clinical studies is based in seven questions. Q. 1: Was the study described as randomized? (Yes: 1 point/No: 0 point); Q. 2: Was the method of randomization described? (Yes: 1 point/No: 0 point); Q. 3: Was the randomization method appropriate? (Yes: 0 point/No −1 point); Q. 4: Was the study described as double blind? (Yes: 1 point/No: 0 point); Q. 5: Was the blinding method described? (Yes: 1 point/No: 0 point); Q. 6: Was the blinding method appropriate? (Yes: 0 point/No: −1 point); Q. 7: Was there a description of withdrawals and dropouts? (Yes: 1 point/No: 0 point). A positive point is awarded in the fourth and six question to the articles even if they were single-blind but well described. In implant placement it is very difficult for the surgeon to proceed if he/she does not know what type of implant placement he/she is performing. In this case, to score the Jadad point, radiographs should be evaluated by a person other than the surgeon who placed the implants. The patients were blinded about which implant was the control or test implant. However, due to the nature of the study, the implant surgeon could not be blinded to the treatment assignment.

**Table 2 materials-12-00154-t002:** Characteristics of studies on randomized control trials (RCT) and non-RCT.

Author	Study Design	N. of Patients	Mean Age of Patients (Years)	N. of Implants	Surgical Technique (One Stage Two Stages)	Implant Insertion Depth below the Crest (mm)	Loading of Implants	Type of Prosthesis	Timing of Implant Placement	Bone Loss in Subcrestal Implants (mm)	Bone Loss in Crestal Implants (mm)	Follow-up (Months)	Jadad Score	Survival Rate (%)	Outcomes Related to Bone Loss between Two Groups
**(a) Characteristics of Studies RCT**															
Pellicier et al. 2016 [6]	RCT	26 15: W 8: M 3: NR	49.8 ± 11.6	23 13: S 10: C	Two stages	S: 2.16 ± 0.88 C:0.0	Delayed loading	Platform switching Crowns screwed	3 months after of the tooth socket healing	1.22 ± 1.06	0.06 ± 1.11	12	4	100	Bone loss was found to be greater in the case of the subcrestal implants
Palaska et al. 2016 [24]	RCT	81 W: 41 M: 40	49	105 54: S 51: C	One stage	S: 1.5 ± 0.2 C: NR	Delayed loading	The implants were not loaded	After a minimum of 3 months of post-extraction healing	Group 1 (Internal connection): 0.68 ± 0.07 Group 3 (Morse taper)0.49 ± 0.06	Group 2 (Internal connection): 0.79 ± 0.06 Group 4 (Morse Taper): 0.40 ± 0.07	3	5	100	No statistically significant difference between implants with the same abutment connection pattern
de Siqueira et al. 2017 [12]	RCT	11	45–65	55 28: S 27: C	Immedia-tely loading	S: 1–3 C: NR	Immedia-tely loading an insertion torque of at least 45 Ncm	Full-arch implant fixed prostheses	NR	1.03 ± 0.60	0.66 ± 0.38	8	5	100	No significant differences
Koutouzis et al. 2014 [26]	RCT	30 W: 24 M: 6	49.85	30 20: S 10: C	Immedia-tely loading	S: −1, −2 C: 0	Immedia-tely loading	Platform-switched screw retained single crowns	Non post-extraction	submerged 1 mm: −0.65 ± 0.45 submerged 2 mm: −0.85 ± 0.75 CBCT analysis	−0.08 ± 0.25	12	4	100	No statistically significant differences between the groups.
Koh et al. 2011 [27]	RCT	24 W: 8 M: 12 (4 patients group up)	55.5	20 10: S 10: C	Two stages	S: 1 (below the palatal crest) C: 0	Delayed loading	NR	Immediate	−0.8 ± 0.6	0.3 ± 0.4	12	4	95.8	No statistically significant differences
Vervaeke et al. 2018 [28]	RCT	25 W: 13 M: 12	65 range = 43–82	50 25: S 25: C	Immediate-ly restored with locator abutment One stage	S: was adapted to the soft tissue thickness, allowing at least 3 mm space for biologic width establish-ment C: NR	Delayed loading	Platform switching Overden-ture	Minimum 3 months post-extraction	0.04	0.73	24	4	100	Subcrestal implants showed significantly better bone levels
Kütan et al. 2015 [29]	RCT	28 W: 21 M: 7	46.05	56 28: S 28: C	Two stages	S: 1 C: NR	Delayed loading	Platform-switched cemented crowns	Minimum 6 months after extraction	1.21 ± 1.05	0.56 ± 0.35	36	5	100	The mean radiographic vertical bone loss in the crestal group was significantly lower than in the subcrestal group
Al Amri et al. 2017 [32]	RCT	23 W: 7 M: 16	43.5	46 23: S 23: C	One stage	S: 2 aprox C: NR	Delayed loading	Screw retained crowns	Healed edentulous	0.3 ± 0.2	0.45 ± 0.2	36	4	100	No significant differences in bone loss around implants placed at crestal and subcrestal levels.
Fickl et al. 2010 [35]	RCT	36 W: 18 M: 18	55.3	89 75: S 14: C	Two stages	S: NR C: NR	Delayed loading	Platform-switched	Healed edentulous	0.30 ± 0.07 at time of insertion of the definitive prosthesis 0.39 ± 0.07 at 1 year	0.68 ± 0.17 at time of insertion of the definitive prosthesis 1.00 ± 0.22 at 1 year	12	3	100	Subcrestal and platform switched implants seem to limit cretsal bone remodeling
**(b) Characteristics of Studies Non-RCT**															
Romanos et al. 2015 [22]	No-RCT Retrospective	85 M: 41 W: 44	50.51	228 197 mesial and distal shoulders: S 65 mesial and distal shoulders: C 194 mesial and distal shoul-ders excluded for being supra-crestal	Two stages	S: at least 0.5 C: within 0.5 mm or less of the crestal bone level) Supracrestal: more than 0.5 mm above the bone level	Delayed loading	Platform-switched Fixed or removable prosthesis	NR	Mesial 1.84 (±1.49) Distal 1.73 (±1.31)	Mesial 1.41 (±1.65) Distal 1.34 (±1.60)	S: 105.61 (±49.74) C: 94.10 (±52.42)	2	97.8	No significant differences
Al Amri et al. 2017 [23]	No-RCT Retrospective	52	45.4 ± 1.8	52 27: S 25: C	One stage	S: 2 mm aprox C: NR	Delayed loading	Single prosthesis. Platform-switched, screw retained metal ceramic	NR	1.2 ± 0.2	1.4 ± 0.2	S: 63.6 ± 2.4 C: 62.4 ± 1.2	2	NR	No significant differences
Nagarajan et al. 2015 [25]	No-RCT Prospective	24	NR 23 to 45	24 12: S 12: C	Two stages	S: 1 C: NR	Delayed loading	NR	Healed edentulous ridges	0.4917 ± 0.4881	0.2183 ± 0.2874	6 (before prosthetic loading)	1	NR	Did not show difference in crestal bone loss before prosthetic loading. Was statistically not significant.
Ercoli et al. 2017 [30]	No-RCT Retrospective	55	57	134 157 mesial and distal shoulders: S 69 mesial and distal shoulders: C 42 mesial and distal shoulders excluded for being supra-crestal	56.6% Two stages 43.4% One stage	NR	Delayed loading	71.7% Single crown 28.3% fixed dental prosthesis	NR	mesial −1.56 ± 1.11 distal −1.06 ± 0.96	mesial −0.72 ± 1.07 distal −0.91 ± 0.83	18	2	100%	No statistically significant differences
Veis et al. 2010 [31]	No-RCT Retrospective	NR	NR	282 89: S 95: C 98: Exclu-ded for being supra-crestal	Two stages and One stage with short healing abutments in the esthetic zone	S: 1 to 2 C: NR	Delayed loading	Cemented an screw crown and ridges	Non post-extraction	Not platform-switching 0.81 ± 0.79 and platform switching 0.39 ± 0.52	Not platform-switching 1.23 ± 0.96 and platform switching 1.13 ± 0.42	24	1	NR	No statistically significant differences. The platform switching concept was beneficial only in subcrestal locations
Degidi et al. 2011 [33]	No-RCT Case series	9	NR	9 4: S 5: C	2: Two stages 7: Immediate-ly loading	S: 1 to 3 C: NR	2 Delayed loading 7 Immediately loading	Platform switching Single crown	NR	Between 0 and 0.5	Between 0.5 and 1.5	1–2	3	100%	In all subcrestally placed implants, preexisting and newly formed bone was found over the implant shoulder
Kim et al. 2017 [34]	No-RCT Retrospective	61	51.4	143 286 implant surfaces 36: S 177: C 73: implant surfaces were excluded for being supra-crestal	Two stages	S:NR C: Within 0–0.75 above the marginal bone level	Delayed loading	No platform-switched	Non post-extraction	1.76 ± 0.78	1.13 ± 0.91	36	2	NR	In subcrestal group signicantly greater bone loss was observed at all time points from the baseline

RCT: randomized control trial; Non-RCT: not randomized control trial; W: woman; M: men; Y: years; NR: not reported; S: subcrestal implants; C: crestal implants; IL: immediately loading; N: number.

**Table 3 materials-12-00154-t003:** Implant characteristics in studies included.

Study	Commercial Brand Surface Neck and Diameters and Lengths in Subcrestal Implants	Commercial Brand, Surface Neck and Diameters and Lengths in Crestal Implants
Romanos et al. 2015 [22]	*Brand implant:* ANKYLOS^®^ Implant System, Mölndal, Sweeden *Type of surface:* Sandblasted, acid-etched with 2 mm of machined collar and a progressive thread design. *Connection:* Internal tapered *Lengths:* NR *Diameters:* NR	*Brand implant:* ANKYLOS^®^ Implant System, Mölndal, Sweeden *Type of surface:* Sandblasted, acid-etched with 2 mm of machined collar and a progressive thread design. *Connection:* Internal tapered *Lengths:* NR *Diameters:* NR
Al Amri et al. (RCT) 2017 [32]	*Brand implant:* Straumann AG, Basel, Switzerland. *Type of surface:* NR *Connection:* Internal connection *Lengths:* 10 to 14 mm. *Diameters:* 3.3 to 4.1 mm	*Brand implant:* Straumann AG, Basel, Switzerland. *Type of surface:* NR *Connection:* Internal connection *Lengths:* 10 to 14 mm. *Diameters:* 3.3 to 4.1 mm
Pellicer et al. 2016 [6]	*Brand implant:* Mozo-Grau^®^ Inhex^®^, S.L. Valladolid, Spain *Type of surface:* NR *Connection:* Internal connection Morse tapered *Lengths:* 10, 11.5, 13 *Diameters:* 3.7, 4.2, 5	*Brand implant:* Mozo-Grau^®^ Inhex^®^, S.L. Valladolid, Spain *Type of surface:* NR *Connection:* Internal connection Morse tapered *Lengths:* 10, 11.5, 13 *Diameters:* 3.7, 4.2, 5
Palaska et al. 2016 [24]	*Brand implant:* Biomet 3i, Palm Beach Gardens, FL, USA Certain Prevail *nanotite.* *Type of surface:* NR *Connection:* Internal connection *Lengths:* 8 to 13 mm *Diameters:* 3.5 to 5 mm	*Brand implant:* OsseoSpeed Astra tech Dental, Molndal, Sweden *Type of surface:* NR *Connection:* Morse tapered *Lengths:* 8 to 13 mm *Diameters:* 3.5 to 5 mm
Nagarajan et al. 2015 [25]	*Brand implant:* ADINT implants. Adin, Co. Afula, Israel *Type of surface:* NR *Connection:* NR *Lengths:* NR *Diameters:* NR	*Brand implant:* ADINT implants. Adin, Co. Afula, Israel *Type of surface:* NR *Connection:* NR *Lengths:* NR *Diameters:* NR
de Siqueira et al. 2017 [12]	*Brand implant:* Titamax CM, Neodent, Curitiba, PR, Brazil *Type of surface:* Sandblasted and acid-etched *Connection:* Internal tapered *Lengths:* NR *Diameters:* NR	*Brand implant:* Titamax CM, Neodent, Curitiba, PR, Brazil *Type of surface:* Sandblasted and acid-etched *Connection:* Internal tapered *Lengths:* NR *Diameters:* NR
Koutouzis et al. 2014 [26]	*Brand implant:* Ankylos CX implants (Dentsply), Mölndal, Sweeden *Type of surface:* NR *Connection:* Morse taper *Lengths:* NR *Diameters:* NR	*Brand implant:* Ankylos CX implants (Dentsply) Mölndal, Sweeden *Type of surface:* NR *Connection:* Morse taper *Lengths:* NR *Diameters:* NR
Koh et al. 2011 [27]	*Brand implant:* Biohorizons, Birmingham, AL, USA. *Type of surface:* with laser-microtextured collar, Laser-Lok *Connection:* Tapered internal *Lengths:* NR *Diameters:* 3.8 or 4.6	*Brand implant:* Biohorizons, Birmingham, AL, USA. *Type of surface:* with laser-microtextured collar, Laser-Lok *Connection:* Tapered internal *Lengths:* NR *Diameters:* 3.8 or 4.6
Vervaeke et al. 2018 [28]	*Brand implant:* Astra Tech Osseospeed TX™, Denstply implants, Salzburg, Austria *Type of surface:* NR *Connection:* Morse taper *Lengths:* 8, 9 or 11 mm *Diameters:* 3.5 or 4 mm	*Brand implant:* Astra Tech Osseospeed TX™, Denstply implants, Salzburg, Austria *Type of surface:* NR *Connection:* Morse taper *Lengths:* 8, 9 or 11 mm *Diameters:* 3.5 or 4 mm
Kütan et al. 2015 [29]	*Brand implant:* Astra Tech Dentsply Implants, Mölndal, Sweeden *Type of surface:* NR *Connection:* Morse taper *Lengths:* 9 mm to 13 mm *Diameters:* 3.5 or 4 mm	*Brand implant:* Astra Tech Dentsply Implants, Astra Tech, Mölndal, Sweeden *Type of surface:* NR *Connection:* Morse taper *Lengths:* 9 mm to 13 mm *Diameters:* 3.5 or 4 mm
Ercoli et al. 2017 [30]	*Brand implant:* NR *Type of surface:* NR *Connection:* NR *Lengths:* NR *Diameters:* NR	*Brand implant:* NR *Type of surface:* NR *Connection:* NR *Lengths:* NR *Diameters:* NR
Veis et al. 2010 [31]	*Brand implant:* full Osseotite implants, Biomet 3i *Type of surface:* NR *Connection:* Screw-type external-hex titanium implants *Lengths:* NR *Diameters:* NR	*Brand implant:* full Osseotite implants, Biomet 3i *Type of surface:* NR *Connection:* Screw-type external-hex titanium implants *Lengths:* NR *Diameters:* NR
Degidi et al. 2011 [33]	*Brand implant:* ANKYLOS plus, DENTSPLY-Friadent, Mannheim, Germany *Type of surface:* Acid-etched microtexturized surface *Connection:* Morse tapered *Lengths:* NR *Diameters:* NR	*Brand implant:* ANKYLOS plus, DENTSPLY-Friadent, Mannheim, Germany *Type of surface:* Acid-etched microtexturized surface *Connection:* Morse tapered *Lengths:* NR *Diameters:* NR
Al Amri et al. 2017 [23]	*Brand implant:* Straumann” Dental Implant System, Institut Straumann, AG Peter Merian-Weg 12 CH- 4002 Basel, Switzerland *Type of surface:* NR *Connection:* Internal connection Regular crossfit connection implants *Lengths:* 10–14 mm *Diameters:* 4.1 mm	*Brand implant:* Straumann” Dental Implant System, Institut Straumann, AG Peter Merian-Weg 12 CH- 4002 Basel, Switzerland *Type of surface:* NR *Connection:* Internal connection Regular crossfit connection implants *Lengths:* 10–14 mm *Diameters:* 4.1 mm
Kim et al. 2017 [34]	*External connection. Brand implant:* Bråemark System MkIII TiUnite, Nobel Biocare AB, Göteborg, Sweden *Type of surface*: 9 TU:TiUnite *Internal connection Brand implant:* OsstemUSII, Osstem Implant Co., Seoul, Korea *Type of surface*: 9 SA: Sand-blasted whit alumina and acid etching, 9 *Brand implant:* Pitt-easy FBR, Oraltronics Dental Implant Technology GmbH, Bremen, Germany *Type of surface*: 18 CP: Calcium phosphate *Connection:* Internal and external *Lengths:* NR *Diameters:* NR	*External connection.Brand implant:* Brånemark System MkIII TiUnite, Nobel Biocare AB, Göteborg, Sweden *Type of surface*: 44TU: TiUnite *Internal connection Brand implant:* OsstemUSII, Osstem Implant Co., Seoul, Korea *Type of surface*: 53 SA: Sand-blasted whit alumina and acid etching, 9 *Brand implant:* Pitt-easy FBR, Oraltronics Dental Implant Technology GmbH, Bremen, Germany *Type of surface*: 80 CP: Calcium phosphate *Connection:* Internal and external *Lengths:* NR *Diameters:* NR
Fickl et al. 2010 [35]	*Brand implant:* Osseotite Certain Biomet 3i, Florida, USA *Type of surface:* Dual acid etched *Connection:* Internal connection *Lengths:* NR *Diameters:* 5 mm	*Brand implant:* Osseotite Certaind Biomet 3i, Florida, USA *Type of surface:* Dual acid etched *Connection:* Internal connection *Lengths:* NR *Diameters:* 4 mm

**Table 4 materials-12-00154-t004:** Soft tissue.

Author	Outcomes Regarding Soft Tissue Analysis in Crestal and Subcrestal Implants
Romanos et al. 2015 [22]	Not Analyzed
Al Amri et al. 2017 [23]	Probing depth: there was no significant difference in both groups Bleeding on probing: there was no significant difference in both groups
Pellicer et al. 2016 [6]	Not Analyzed
Palaska et al. 2016 [24]	Probing depth: was deeper in subcrestal positions but there was no statistically significant difference between groups (*p* > 0.05) Biotype: no statistically significant difference was recorded between groups Modified plaque index: no statistically significant difference between crestal and subcrestal implants Modified gingival index: highest values were recorded for crestal group, being statistically significant from subcretsal group
Nagarajan et al. 2015 [25]	Not Analyzed
de Siqueira et al. 2017 [12]	Keratinized tissue width: no differences in two groups Vertical mucosa thickness: no differences in two groups Tissue recession: no differences in two groups Plaque index bleeding on probing: peri-implant tissues were checked and found healthy at every follow-up return in the two groups
Koutouzis et al. 2014 [26]	Not Analyzed
Koh et al. 2011 [27]	Keratinized tissue width: from baseline to 4 months was 0.7 ± 0.2 mm for the crestal group and 1.7 ± 0.4 mm for the subcrestal group being statistically significantly
Vervaeke et al. 2018 [28]	A significant correlation was observed between soft tissue thickness and bone level alterations after 6 months with inferior bone levels for crestal implants when thin tissues are present
Kütan et al. 2015 [29]	Probing depth: no significant differences Modified plaque index: no significant differences Modified gingival index: no significant differences
Ercoli et al. 2017 [30]	Not Analyzed
Veis et al. 2010 [31]	Not Analyzed
Degidi et al. 2011 [33]	Histological analysis Crestal: dense connective tissue, with only a few inflammatory cells, was observed at the level of the shoulder of the implant and of the periimplant coronal portion Subcrestal: no gaps or fibrous connective tissue was found at the bone-implant interface.
Al Amri et al. 2017 (RCT) [32]	Probing depth: no significant differences Bleeding on probing: no significant differences
Kim et al. 2017 [34]	Not Analyzed

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
