# Peer review of "Bone Loss in Implants Placed at Subcrestal and Crestal Level: A Systematic Review and Meta-Analysis"

_materials, 2019, doi:10.3390/ma12010154_

Round 1
Reviewer 1 Report
Taking into account all the several features, the accuracy, scientific quality, scientific content and the interpretation of the results are very good.
- The approach is interesting and the topic is appropriate for the journal.
- The work has a very clear structure and all the sections are well written in a way that is easy to read and understand. In addition, the structure of the paper is very good.
- “Table 2a & 2b” should be replaced by “Tables 2a and 2b”. The authors should check it.
- It seems that the paper does not contain repetitions.
- The quality of some figures should be improved.
- The title is adequate and appropriate for the content of the article.
- The abstract contains information of the article.
- Figures and captions are essential and clearly reported.
Author Response
Response to the reviewer's comments in Word file: Reviewer 1

Reviewer 2 Report
The present study analyses the effect of crystal v/s subcrestal placement of implants on bone loss and other soft tissue parameters. The study has been well designed and written. I have a few concerns as mentioned below:
The first concern with the study is the lack of inclusion/exclusion criteria for the follow-up period. Two of the included studies have follow-ups of 1-2 and 3 months, respectively. It would be good to include studies with a follow-up period of at least 6 months, preferable 12 months, post loading. The authors have done well by not including these studies in the meta-analyses, but it would be good to have strict inclusion criteria in regards to the follow-up period for the benefit of the readers.
The second concern is regarding the meta-analysis for non-RCT studies. You have included 2 studies, one having implants with ext connection and the other with both ext and int connection. In my opinion, it is something which, ideally, should not be compared. It would be a good idea to not include this analysis in the article.
It would be good to see the effect of immediate v/s delayed loading separately to find if the timing of loading has an effect on the bone loss in both groups.
Some other minor modifications are listed below:
Abstract
Pg 1 lines 15,16 - Please mention both the start and end period of the literature search.
Pg 1 line 21 - Please avoid starting a sentence with a numeral eg. replace "3" with "Three". Please make sure to make modifications like this in the entire manuscript.
Pg 1 line 24 - Avoid using the word "similar" unless the values are exactly, numerically, similar. You could rephrase it as "comparable" or you could rephrase the whole sentence.
Introduction
Pg 1 lines 43,44 - "Thus, with the...achieving secondary stability". Please rephrase the sentence in a grammatically correct way or join it with the previous sentence.
Pg 2 line 74 - Incomplete sentence.
Results
Pg 6 line 173 - "With respect to the moment of implant placement", please replace 'moment' with 'timing'.
Pg 12 line 189 - Replace 'RTC' with 'RCT'.
Author Response
Response to the reviewer's comments in Word file: Reviewer 2

Reviewer 3 Report
Authors shold improve the English grammar
Author Response
Response to the reviewer's comments in Word file: Reviewer 3
